# Comparative Analysis of Drought Indices in Hydrological Monitoring in Ceará's Semi-Arid Basins, Brazil

Suellen Teixeira Nobre Gonçalves [1,2,*], Francisco das Chagas Vasconcelos Júnior [3], Cleiton da Silva Silveira [1], Daniel Antônio Camelo Cid [1], Eduardo Sávio Passos Rodrigues Martins [3] and José Micael Ferreira da Costa [1]

1. Civil Engineering Department, Campus do Pici, Federal University of Ceará (UFC), Fortaleza 60400-900, CE, Brazil
2. Master's Program in Climatology and Applications in CPLP Countries and Africa, Ceará State University (UECE), Fortaleza 60714-130, CE, Brazil
3. Ceará Institute for Meteorology and Water Resources (FUNCEME), Fortaleza 60115-221, CE, Brazil
* Correspondence: suellen.nobre@gmail.com

**Abstract:** The purpose of this work is to evaluate the applicability of five drought indices (Surface Water Supply Index—SWSI, Reclamation Drought Index—RDI, Streamflow Drought Index—SDI, Standardized Precipitation Index—SPI, and Evaporative Demand Drought Index—EDDI) as tools for monitoring, in identifying the duration, intensity, and frequency of hydrological droughts in the basins of the Banabuiú, Castanhão, and Orós reservoirs, located in the state of Ceará, Brazil. The analysis focused on determining the performance of the indexes for capturing the droughts characteristics from 2002 to 2020. Thus, the comparison of the values of the indexes with the Target Levels of the reservoir's operation were used, as well as the analysis of six decision criteria: robustness, tractability, transparency, sophistication, extensibility, and dimensionality, to compare the behavior of drought indices. The results of the evaluation criteria showed that the SPI was superior to the other indices, being able to significantly represent the drought episodes, capturing a greater number of events. Thus, SPI received the highest total weighted score (118), followed by SWSI (97), EDDI (95), SDI (95), and RDI (88). In this context, it was found that the SPI and SWSI are the most suitable indices to monitor drought in the region and that the use of longer time scales can be recommended to manage hydrological droughts, in order to improve planning and management of water resources.

**Keywords:** drought indices; reservoir; monitoring



## 1. Introduction

Northeast Brazil (NEB) is a location with high vulnerability to climatic factors, with its semi-arid area being the most susceptible. Among the elements that constitute the climate, precipitation has a strong influence on agricultural activities, since, in the tropics, the rainfall regime tends to be of short duration and high intensity. In addition, it acts directly in modifying the landscape and the environment [1–3].

The state of Ceará, as well as other regions of NEB, presents temporal and spatial variations of precipitation, high temperatures, and high evaporation rates. The accumulated precipitation in this region can vary from 250 to 800 mm.year$^{-1}$ and be distributed among the first five months of the year, which generates a concentration of rainfall in some locations and scarcity in others. The irregularity in the rainfall causes periodic droughts, which in recent years have been recorded more frequently. Climate change can increase the occurrence of droughts, prolonging their duration and intensity, which could directly impact the quality and availability of water in the region [4–6].

The cycle of droughts, which began in the year 2010 and lasted until the year 2019, can be considered the most critical in the last 50 years, as it affected more than 1500 municipalities in the region, leading them to decree a state of emergency. During this period, the Castanhão, Orós, and Banabuiú reservoirs were in a state of hydrological drought, with

their volumes below 20%. With hydric scarcity, there were direct effects on the activities that use these resources, damaging economic, social, and environmental impacts for the affected regions. In addition, the supply of some communities is now carried out by water tankers. However, many places—remote and difficult to access—were left without water and alternative sources, further aggravating the situation of hunger and poverty of those people [1,4,7].

Although NEB is marked by a long history of crisis management policies (building dams and cisterns, well drilling, and financial aid), it is possible to improve the monitoring of the water resources, predictive systems, and early warning, through the use and improvement of instruments used in drought management, such as drought indices. In addition, it is also important to conduct studies of environmental, social, and economic impacts and enable mitigation measures and planning [1,2,7].

Given the background, drought indices are support tools that aim to provide information on the severity, duration, and frequency of drought events. In this way, they can be used to (a) generate an early warning system for droughts; (b) assess fire and storm risks; (c) calculate the probability of ending a drought event; (d) analyze temporally and spatially the characteristics of drought, comparing different regions; and (e) assist in public policies for assistance to the effects of drought [8].

There is a large number of drought indices in the literature; however, some of them have restricted utility, since their calculation and interpretation are limiting factors for their use. In this case, the lack of input data, failures in the historical series, high computational demand, and the need for qualified professionals are factors that may restrict the use of a particular index. Furthermore, many indices fail to capture drought events and end up performing poor quality monitoring [9,10].

Thus, some indices are more appropriate than others and this depends on the type of drought that one may wish to monitor. In the case of hydrological drought monitoring, one can cite the use of the Surface Water Supply Index (SWSI), Reclamation Drought Index (RDI), Streamflow Drought Index (SDI), and Evaporative Demand Drought Index (EDDI). For the monitoring of meteorological drought, the following stands out: the Standardized Precipitation Index (SPI) [11–15].

Several studies have already been conducted to evaluate the ability of the indices to identify and characterize drought events. The authors of [16,17] conducted a comparative analysis of five drought indices based on the decision criteria (robustness, treatability, transparency, sophistication, and extensibility) to identify the most appropriate indexes for monitoring drought events in the study region. The authors of [16] evaluated the performance of the indices (Percent Departure from Normal (PDN), Effective Drought Index (EDI), SPI, Reclamation Drought Index (RDI), and Standardized Precipitation Evapotranspiration Index (SPEI)) for a semi-arid basin, located in the western region of India. The period analyzed was 25 years (1985–2009) and the results indicated that the 9-month scale is the most appropriate for comparing the drought indices under study. Additionally, the authors concluded that the SPEI-9 presented a better identification of droughts and a higher score concerning the decision criteria. Therefore, it was indicated by the authors as the most appropriate index to carry out the monitoring of meteorological drought in the studied region. The researchers of [17] analyzed the capacity of the indices Percentage of Normal (PN), Decis, SPI, SWSI, and Aggregate Drought Index (ADI) in modeling the historical droughts that have occurred in the Yarra River basin in the state of Victoria, Australia. From the evaluation of the decision criteria, the authors found that the ADI proved to be superior to the other indices, being indicated for drought management in the studied area.

In Brazil, some papers addressed this issue, as in the studies conducted by [18,19]. The authors of [18] compared the performance of the indices (SPI, SPEI, Standardized Runoff Index—SRI, State Index—SI, Synthetic Index, and the Target Levels—TL), through qualitative and quantitative analysis, for the Jucazinho reservoir, located in Pernambuco. Consequently, it was possible to conclude that the severity of drought is not always related

to natural factors and that the SPI, SPEI, and SRI indexes present difficulties in tracking fluctuations in the availability of water from the reservoir.

The research carried out by [19] comparatively analyzed the SPI and the Palmer Drought Index (PDSI, in original form and adapted to the climatic conditions of the state of São Paulo). Hence, the authors pointed out that the SPI presents usage versatility, temporal analyses consistency, and simple understanding, and can be applied in drought monitoring. In addition, the PDSI (adapted) also showed that it was effective in quantifying the meteorological drought for the region studied.

Although the Brazilian literature is replete with works referring to drought indices, when one makes further analysis on the use of the decision criteria for benchmarking these tools, there is a lack of studies that address this comparison for watersheds. Moreover, it is noted the importance of conducting specific research for the context of the region to be studied, because the effectiveness and dexterity of the indices vary according to their applicability in a given locality.

In light of the above, the present article aims to evaluate the drought indices SWSI, RDI, SPI, SDI, and EDDI, in monitoring the hydrological drought of the Banabuiú, Castanhão, and Orós reservoirs, located in the state of Ceará, Brazil. This evaluation was based on the analysis of the droughts that occurred in the study region, as well as on the comparison of index values about the Target Levels of operation and the six decision criteria (robustness, treatability, sophistication, transparency, extensibility, and dimensionality) proposed by [20].

## 2. Materials and Methods

The methodology of the study encompassed six main steps, as follows:

(1) Identification of the study area, in which the reservoirs of greatest water importance for the states of Ceará, Castanhão, Orós, and Banabuiú;
(2) Analysis of the time series of precipitation, flow, reservoir level, temperature, and potential evapotranspiration;
(3) Calculation of drought indexes (SWSI, RDI, SDI, SPI, and EDDI);
(4) Definition of the values of the reservoir operation Target Levels;
(5) Quantification of the droughts that occurred in the study region according to the Target Levels;
(6) Analysis of six decision criteria.

Figure 1 presents the flowchart of the processes involved in the construction of the work.

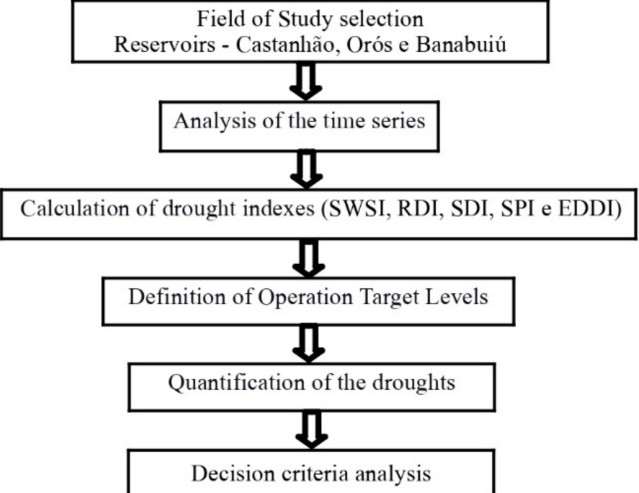

**Figure 1.** Diagram of the methodology used.

*2.1. Study Area*

Reservoirs, also known as dams, weirs, and barrages, are systems that are formed by anthropic action, through the damming of water bodies. Their function is to accumulate water from the rainy season and store it for dry periods. From this form, depending on the objective and purpose of creation, its reserves can be earmarked for the water supply, power generation, irrigation, regularization of the natural flow of the barred body, navigation, recreational and socioeconomic activities [21–24].

The purpose of monitoring weirs is to accompany their water levels and streamflow (affluent and effluent). This information is important because it can help in the decision-making process related to water operation, thus allowing a better use and exploitation of this resource [21].

Therefore, the dams selected for the study were Banabuiú, Castanhão, and Orós. The three reservoirs (Figure 2) located in the semi-arid region of the state of Ceará are part of the Jaguaribe–Metropolitano and were built on the Banabuiú, Médio Jaguaribe, and Alto Jaguaribe, respectively. The Castanhão dam has a maximum capacity of 6.7 billion m$^3$, followed by Orós with 1.94 billion m$^3$ and Banabuiú with 1.6 billion m$^3$. Together, these reservoirs add up to an accumulation capacity of 10,241,000 hm$^3$ and perennial stretches of 150.34 km (Castanhão), 109.24 km (Orós), and 135.90 km (Banabuiú).

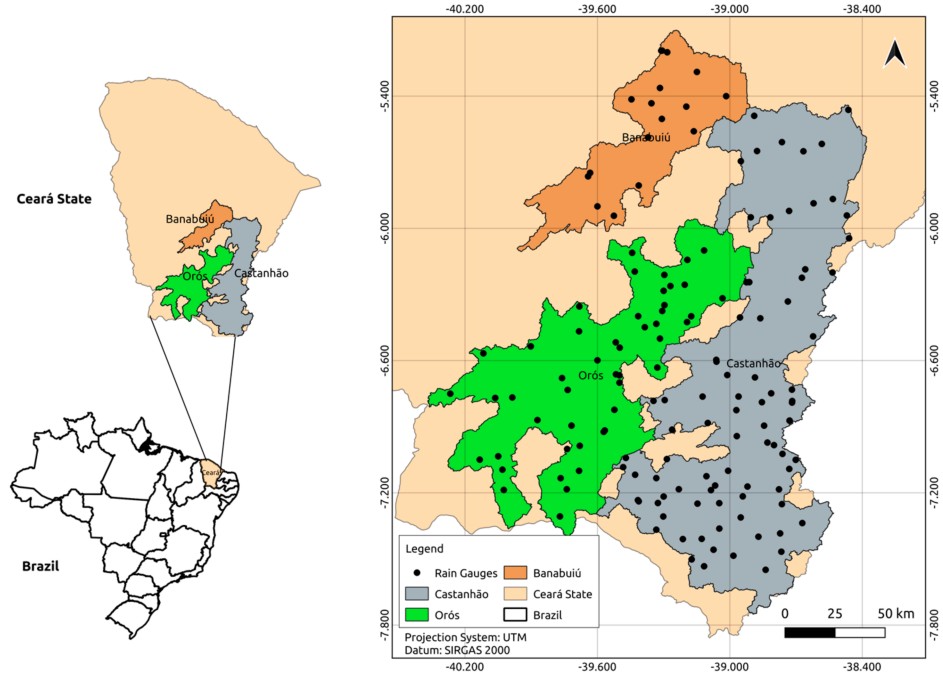

**Figure 2.** Study basins and location of rain gauges. Number of rain gauges in the Banabuiú, Castanhão, and Orós basin are 17, 83, and 50, respectively.

They are considered important water sources because they are strategic reservoirs for the state, where one of its main attributions is the supply of water to the Metropolitan Region of Fortaleza (MRF). The RMF is the sixth most populous region in Brazil with a population of over 4 million inhabitants [25].

*2.2. Data Used*

The data used in this work (Table 1) to calculate the drought indices were collected from three institutions, namely: (a) Ceará Institute for Meteorology and Water Resources (FUNCEME) [26–28], which provided data on precipitation (Figure 2), streamflow, and potential evapotranspiration (PET) for each reservoir's drainage system; (b) Ceará Water Resources Management Company (COGERH) [29], which granted the data of levels and

volumes of the reservoirs; and (c) National Institute of Meteorology (INMET) [30], which provided temperature data from the municipalities where each reservoir is located.

**Table 1.** Data used in the construction of the work.

| Variables | Institution | Period |
|---|---|---|
| Precipitation | FUNCEME's raingauge stations | 1980–2020 |
| Streamflow | Rainfall-flow Hydrologic Model MODHAC (FUNCEME) | 1980–2020 |
| Level and Volume | Hydrological Portal (COGERH) | Castanhão (2002–2020) Orós (1986–2020) Banabuiú (1986–2020) |
| Temperature | Meteorological Stations (INMET) | 1980–2020 |
| Evapotranspiration | Hargreaves estimation (FUNCEME—Hargreaves; Samani 1985) | 1980–2020 |

Thiessen polygon technique [31] was used to calculate the mean precipitation and PET over the reservoir's drainage basin area. In addition, in the absence of temperature data for the municipalities of the reservoir's origins, it was then adopted the data corresponding to the regions closest to the area of interest.

*2.3. Drought Indices*

2.3.1. SWSI

The SWSI was developed by [15]. It was used in studies on the periods of droughts and flooding in basins, representing an index with predictive potential, which associates meteorological (precipitation) and hydrological (streamflow and reservoir level) measures into a numerical value. Thus, it is possible to monitor abnormalities in the surface water supply [10,15,32–37].

The calculation of the SWSI (Equation (1)) can be performed by obtaining monthly data of precipitation, streamflow, and reservoir level, which must be normalized through an analysis of cumulative frequency, and then subsequently obtaining the probabilities of non-exceedance for each variable. A monthly aggregation of the values of each component is then carried out, with the assignment of weights, which will depend on the contribution of each variable to the surface water supply in each basin. Immediately after, the weighted variables are summed, in order to establish an SWSI value that can represent the whole basin. Subtraction by 50 and subsequent division by 12 are operations performed to make the SWSI scale similar to the Palmer *Drought Severity* Index (*Palmer Drought Index* PDSI) [15,32,37].

$$\text{SWSI} = \frac{\left( \left[ \left( a \cdot P_{precipitation} \right) + \left( b \cdot P_{streamflow} \right) + \left( c \cdot P_{level} \right) \right] - 50 \right)}{12} \quad (1)$$

where:

- $a$, $b$, and $c$ are the coefficients for precipitation over reservoir drainage area, inflow into the reservoir, and reservoir level, respectively, and must meet the condition: $a + b + c = 1$;
- $Pi$ is the probability (%) of non-exceedance for each of these components.

Weighting Factors

Each input variable to the SWSI (precipitation, streamflow, and level) has a different weighting to the index. For example, the inflow into a reservoir has more relevance to water availability than the rainfall over some point in the watershed, since there are losses other than evaporation when runoff occurs. This is also true for the level of the reservoir, as it directly informs the amount of water stored at a given moment, while the other two

(precipitation and streamflow rate) indicate the potential for water storage, i.e., they have indirect relevance.

Based on this principle, weighting values were assigned to the coefficients associated with rainfall, streamflow, and reservoir level, in an attempt to express the contribution of each input data. Thus, a value of 1 was assigned for precipitation, 2 for streamflow, and 3 for reservoir level.

It is important to emphasize that the weighting factors are not inserted in the original equation of the index (Equation (1)). This insertion was made to adapt the Northeast of Brazil, in light of the different climatic conditions of the region.

Adaptation of the SWSI Classification

Table 2 shows an adaptation of the SWSI index classification for NEB in order to seek a better nomenclature for each category presented. It is worth noting that the values obtained for the index are associated with the availability of water in reservoirs and a potential tendency for drought or water abundance events.

**Table 2.** SWSI classification of water availability in reservoirs.

| SWSI | Classification |
|---|---|
| 3.00 to 4.00 | Abundant Supply (SUPA) |
| 2.00 to 2.99 | Moderate Supply (SUPM) |
| 1.00 to 1.99 | Above normal Supply (SUPAN) |
| −0.99 to 0.99 | Normal (N) |
| −1.99 to −1.00 | Initial Drought (ID) |
| −2.494 to −2.00 | Moderate Drought (MD) |
| −2.99 to −2.495 | Severe Drought (SD) |
| −4.00 to −3.00 | Extreme Drought (ED) |

Note: Source: Adapted from [15].

When analyzing the original classification of the SWSI, it can be seen that it does not include the Severe Drought category. The lack of this classification can directly impact the index analysis since the Target Levels (Topic 2.8) use four categories to identify droughts. Thus, we added the Severe Drought classification to the SWSI, which was obtained from the average values between Moderate Drought and Extreme Drought. This insertion aimed to improve the index analysis and make it comparable to the others.

2.3.2. RDI

RDI was formulated by [14] in Denver, Colorado, to clarify climatic and water supply factors for that locality. It is based on precipitation, streamflow, reservoir level, and temperature values, and can be used for drought monitoring and early warning purposes. RDI is also calculated at the basin level so that its scale resembles that of the SWSI.

RDI calculation (Equation (2)) uses monthly precipitation, streamflow, reservoir level, and temperature data. Similar to the SWSI, each variable will be normalized using a cumulative frequency distribution from a historical data series. The weight of each component will be assigned according to its contribution and influence on that basin. The probability of non-exceedance will be calculated for all variables and then these values will be multiplied by 100 and subtracted from 50. The result value can be positive and negative values, ranging from −50 to 50, and is defined as the monthly index of non-exceedance values [14].

Moving averages of twelve months were calculated for the monthly index of non-exceedance values, which will correspond to the averages between the current month and the previous eleven months. Obtaining these values makes it possible to identify drought patterns and provides important information for monitoring. The duration factor of the variables is obtained by counting the months that have positive values for the monthly non-exceedance values and the twelve-month moving average. If both are negative, the duration factor will correspond to the number of negative months. In the last case, when

one is positive and the other negative, the duration factor will be equal to the minimum value of 6 [14].

$$
\begin{aligned}
\text{RDI} = {} & \frac{a}{60} \times (12 - monthmovingaverage\ \text{Prec}) \times (DurationFactor\ \text{Prec})^{1,2} \\
& + \frac{b}{60} \times (12 - monthmovingaverage\ Flow) \times (DurationFactor\ Flow)^{1,2} \\
& + \frac{c}{60} \times (12 - monthmovingaverage\ Level) \times (DurationFactor\ Level)^{1,2} \\
& + \frac{d}{60} \times (12 - monthmovingaverage\ Temp) \times (DurationFactor\ Temp)^{1,2}
\end{aligned}
\tag{2}
$$

where:

- $a$, $b$, $c$, and $d$ are the coefficients for precipitation over the reservoir drainage area, the reservoir inflow, reservoir level, and temperature, respectively, and must meet the condition: $a + b + c + d = 0.5$;
- 12-month moving average is the average of the current month with the previous eleven months, where *Prec* = Precipitation and *Temp* = Temperature;
- Duration factor is the number of positive or negative months, obtained from the count of the monthly Index of non-exceedance values and the twelve-month moving average.

As in the SWSI, weighting factors were also established for the RDI, as a way to adapt it to the study region. Thus, the same values were kept for precipitation (1), streamflow (2), and level (3), but the temperature variable was incorporated and given a value of 2.

The RDI classification of water availability in the region follows the same pattern as the SWSI and can be found in Table 2.

### 2.3.3. SDI

The SDI was proposed by [13], to characterize hydrological droughts occurring in the Evinos River basin in Greece. In performing this study, the authors were able to demonstrate the effectiveness of the SDI in characterizing hydrological drought events, suggesting the application of this index on a global scale. It is considered an index of easy understanding and its calculation was based on the SPI, differing by the input data included, i.e., it uses the average streamflow instead of precipitation [9,13,38–40].

To obtain the SDI (Equation (3)), the collected streamflow data must be fitted to probability functions (normal, log-normal, and gamma). Thus, the index is determined based on the accumulated volume of flows for each period of the hydrological year [13].

$$
\text{SDI} = \frac{V_{i,k} - \overline{V}_k}{S_k} \quad k = 1, 2, 3, \ldots \ i = 1, 2, 3, \ldots
\tag{3}
$$

where:

- $V_{i,k}$ is the accumulated value for the period;
- $\overline{V}_k$ is the average for the selected period from the historical series;
- $S_k$ is the standard deviation for the selected period from the historical series.

Classification of the SDI Index

Positive SDI values correspond to wet conditions, while negative values indicate hydrological drought. Table 3 presents the classification of the SDI, based on its values.

**Table 3.** SDI classification.

| State of Drought | SDI | Classification |
|:---:|:---:|:---:|
| 0 | $\geq 0$ | No Drought |
| 1 | $-1.00 \leq$ and $<0$ | Mild Drought |
| 2 | $-1.5 \leq$ and $<-1.00$ | Moderate Drought |
| 3 | $-2.00 \leq$ and $<-1.5$ | Severe Drought |
| 4 | $<-2.00$ | Extreme Drought |

Note: Source: [13].

### 2.3.4. SPI

The SPI was conceived by [41] in order to quantify precipitation deficit on various time scales and aid in the assessment of drought severity. Quarterly and four-month scales relate to the identification of short-term (meteorological) droughts, while longer scales (12, 24, and 36 months) tend to identify long-term (hydrological) droughts [9,10,32,41–44].

To perform the SPI calculation (Equations (4) and (5)), monthly accumulated precipitation data must be fitted to a time frame of n months. Then, the gamma function is applied to these series, with the subsequent generation of probabilities of precipitation values occurrence. Then, the inverse of the normal distribution is used to find the deviations of precipitation from the mean values of the examined intervals [32,41].

$$\text{SPI} = -\frac{t - C_0 + C_1 t + C_2 t^2}{1 + d_1 t + d_2 t^2 + d_3 t^3} \quad If \ 0 < H(x) \leq 0.5 \tag{4}$$

$$\text{SPI} = \frac{t - C_0 + C_1 t + C_2 t^2}{1 + d_1 t + d_2 t^2 + d_3 t^3} \quad If \ 0.5 < H(x) \leq 1 \tag{5}$$

where:

- $t = \sqrt{\ln\left[\frac{1}{(H(x))2}\right]}$ for $0 < H(x) \leq 0.5$ and $t = \sqrt{\ln\left[\frac{1}{(1-(H(x))2)}\right]}$ for $0.5 < H(x) \leq 1$;
- $H(x)$ = cumulative probability distribution;
- $C_0$, $C_1$, $C_2$, $d_1$, $d_2$, and $d_3$ are constants and equal, respectively, to the values: 2.515517; 0.802853; 0.010328; 1.432788; 0.189269; and 0.001308.

### SPI Classification

The SPI classification is illustrated in Table 4.

**Table 4.** SPI Classification.

| SPI | Classification |
|:---:|:---:|
| $\geq 2.00$ | Extreme Rainfall |
| 1.49 to 1.99 | Severe Rainfall |
| 0.99 to 1.49 | Moderate Rainfall |
| 0.49 to 0.99 | Weak Rainfall |
| $-0.49$ to 0.49 | Almost Normal |
| $-0.99$ to $-0.49$ | Mild Drought |
| $-1.49$ to $-1.00$ | Moderate Drought |
| $-1.99$ to $-1.50$ | Severa Drought |
| $\leq -2.00$ | Extreme Drought |

Note: Source: [42].

### 2.3.5. EDDI

EDDI was developed by [12,45] as an agricultural and hydrological drought monitoring tool. In addition, it has predictive potential and provides warnings about wildfire risks. This index examines evaporative demand anomalies of the atmosphere, which can lead to drought situations in soils and vegetation. It is a multiscale index, meaning that it can be

calculated in several time windows, so that its analysis period can vary, in order to capture drought dynamics operating at different time scales [12,45–47].

The EDDI calculation (Equation (6)) uses potential evapotranspiration data, which can be aggregated according to the selected time scale. Then, the empirical probability is employed, in order to compare each reference month of the series with the complete series of that month. In this way, it is possible to obtain the positional value of the month in relation to the year, thus generating a probability series for each reference value, according to Tukey's empirical plotting position (Wilks [48]. After these statistical analyses, each $P(E_{0_i})$ value is approximated by the inverse of the normal distribution [12,49].

$$\text{EDDI} = W - \frac{C_0 + C_1 W + C_2 W^2}{1 + d_1 W + d_2 W^2 + d_3 W^3} \tag{6}$$

where:

- $P(E_{0_i}) \leq 0.5$, $W = \sqrt{-2\ln\left[P(E_{0_i})\right]}$ and $P(E_{0_i}) \leq 0.5$, replace $P(E_{0_i})$ with $\left[1 - P(E_{0_i})\right]$;
- $C_0$, $C_1$, $C_2$, $d_1$, $d_2$, and $d_3$ are constants and equal, respectively, the values: 2.515517; 0.802853; 0.010328; 1.432788; 0.189269; and 0.001308.

Classification of the EDDI index

Values of EDDI equal to zero indicate that the accumulated $E\_(0\_i)$ in the aggregation period in the year and month of interest is equal to the median value of the series. Negative values indicate a humid anomaly and positive values indicate a drier-than-normal condition. Thus, the higher the positive value, the greater the intensity of the drought indicated by the index (Hobbins et al., 2016). Table 5 shows the EDDI classification.

**Table 5.** EDDI Classification.

| State of Drought | EDDI | Classification |
| --- | --- | --- |
| ED4 | ≥2.00 | Extreme Drought |
| ED3 | 1.49 to 1.99 | Severe Drought |
| ED2 | 0.99 to 1.49 | Moderate Drought |
| ED1 | 0.49 to 0.99 | Mild Drought |
| ED0 | −0.49 to 0.49 | Normal |
| EW1 | −0.99 to −0.49 | Weak Humidity |
| EW2 | −1.49 to −1.00 | Moderate Humidity |
| EW3 | −1.99 to −1.50 | Severe Humidity |
| EW4 | ≤−2.00 | Extreme Humidity |

Note: Source: [50].

*2.4. Target Levels*

The operation of a reservoir consists in defining the water volumes that should be released or stored, within a period of time, in order to meet the demand for this asset. This practice represents a decision-making method that counts on the participation of social, political, and economic agents and presents itself as a relevant phase to increase the efficiency of water resources management [51,52].

The operation of reservoirs is a process that presents uncertainties; therefore, it is important that the decision maker uses tools that are suitable to perform the analyses and that serve as guidance for his choices, such as drought indices. In addition, it is worth noting that if there are operating errors in the system, this can directly influence the evaluation of drought severity by the indices [51,53].

Hence, regions that frequently experience drought periods generally adopt policies to restrict water release based on reservoir stocks [54]. These operating policies are called "hedging" or safeguards and consist of setting small restrictions over time, with the goal of preventing reservoirs from reaching water collapse.

Cid [51], used this approach to define an optimal operation policy applied to drought management in the Jaguaribe—Metropolitan Reservoir System. The reservoir operation policy, named by the author as Target Levels (percentage of the Target Volume related to the total reservoir volume), was built through collaborative modeling with the decision makers of the studied region, incorporating a preference system of these users to a reservoir optimization model.

The reference values of the Target Volumes (Table 6) used in this paper were taken from [51] and correspond to the Long-Term Operation Rule—Target Levels 4, being the most indicated by the author for usage in the reservoirs under study. Thus, the values for the month of December were considered since annual scales were adopted for the calculation of drought indexes.

**Table 6.** Coefficients of the Target Volume.

| Month | Target Volumes | | | |
| --- | --- | --- | --- | --- |
| | Target Volume 1 | Target Volume 2 | Target Volume 3 | Target Volume 4 |
| December | 0.52 | 0.33 | 0.21 | 0.10 |

Note: Source: Adapted from [51].

Based on the values presented in Table 6, the calculations of the Target Levels for the Castanhão, Orós, and Banabuiú reservoirs were performed. For such, the coefficients of the Target Volumes curves presented by [51] were multiplied one by one by the total volumes of each system analyzed, i.e., for Castanhão, the coefficients of the four Target Volumes curves were multiplied by its total capacity (6700 hm$^3$); in the same way, it was conducted for Orós (1940 hm$^3$) and Banabuiú (1200 hm$^3$). Therefore, it was possible to obtain the new values of specific meta volumes for each reservoir analyzed and, from this, to define the drought categories according to their hydric state.

Table 7 presents the classifications of the Target Levels according to the drought category. Thus, Target Level 1 corresponds to the normality situation and is represented by light blue color. Target Level 2 represents an alert when drought begins in the reservoir, and its color corresponds to yellow. Target Level 3 indicates drought, i.e., a situation in which there is an ongoing drought, and its color is orange. For Target Level 4, there is severe drought, which is represented by the color red. Lastly, Target Level 5 corresponds to extreme drought and is related by the color dark red.

**Table 7.** Classification of the Target Levels according to drought categories.

| Target Level | Drought Category |
| --- | --- |
| 1 | Normality |
| 2 | Alert (Initial Drought) |
| 3 | Drought |
| 4 | Severe Drought |
| 5 | Extreme Drought |

Note: Source: Adapted from [51].

*2.5. Quantification of Drought Events According to the Target Levels*

We assumed that each year would correspond to an event within the analyzed period (2002–2020). Thus, the Banabuiú and Castanhão reservoirs (Table 8) presented eight years of hydric normality (Target Level 1), whereas Orós had seven years of normality. In this same context, for Target Level 2, the reservoirs Banabuiú and Castanhão had two events, while Orós had four events. For Target Level 3, Banabuiú had two events and Castanhão and Orós had one event. Target Level 4 had one event in Banabuiú, two events in Castanhão, and four events in Orós. Target Level 5 presented six events in Banabuiú and Castanhão, and only three in Orós.

**Table 8.** Analysis of droughts in relation to the Target Levels for the Banabuiú, Castanhão, and Orós reservoirs.

| Target Level | Quantity of Events | | |
|---|---|---|---|
| | **Banabuiú** | **Castanhão** | **Orós** |
| 1 (Normality) | 8 | 8 | 7 |
| 2 (Alert) | 2 | 2 | 4 |
| 3 (Drought) | 2 | 1 | 1 |
| 4 (Severe Drought) | 1 | 2 | 4 |
| 5 (Extreme Drought) | 6 | 6 | 3 |
| Total of Drought Events | 11 | 11 | 12 |

From this analysis, it was evident that for the 19-year period, 11 drought events occurred in Banabuiú and Castanhão, and 12 events in Orós, ranging from initial to severe.

*2.6. Analysis of Decision Criteria*

The weights were assigned to the six decision criteria (robustness, treatability, transparency, sophistication, extensibility, and dimensionality) based on the literature review [8,16,17,20,55] and analysis related to the study region.

The criterion of robustness was listed as the most important, as it directly reflects on the reliability and identification of drought, so it was assigned the maximum weight (27%). This is followed by sophistication (20%), treatability (20%), transparency (17%), extensibility (10%), and dimensionality (6%), according to Table 9.

**Table 9.** Decision criteria.

| Criteria | Theoretical Definition | Weights | Relative Importance (%) |
|---|---|---|---|
| Robustness | The ability of a technique not to vary with minor deviations. This criterion allows the index to identify physical changes in the environment. In addition, robustness demonstrates the index's ability to identify drought over time under a variety of conditions. This criterion also refers to the index's ability to be compared spatially and temporally, i.e., an index calculated for the northern region of Ceará can be compared directly with an index calculated for another region of the state. Robust indices also present values independent of seasonality (values for one season can be compared directly to other seasons of the year) and should be sensitive to the impacts of drought. | 8 | 27 |
| Sophistication | Refers to the accuracy with which the index is evaluated, so even if an index is not transparent, it will be valuable if it correctly presents the important physical aspects of drought. | 6 | 20 |
| Treatability | Demonstrates the practicality of the index; so, treatable indexes have simple calculations, fewer input variables, and an extensive database available. | 6 | 20 |
| Transparency | Is based on the clarity, rationality, and justification of the index, in order to determine whether decision makers and society can easily understand the methodology employed in its construction. | 5 | 17 |

**Table 9.** *Cont.*

| Criteria | Theoretical Definition | Weights | Relative Importance (%) |
|---|---|---|---|
| Extensibility | Involves the expansion degree of the index over the years, to reflect drought events that have occurred at different periods in history. | 3 | 10 |
| Dimensionality | Displays the connection between the index and the physical medium, in order to investigate whether the tool used is capable of representing fundamental units of the measurement system or only a fraction of them. | 2 | 6 |

In addition, raw scores ranging from 1 to 5 were assigned to all six decision criteria, based on the quantitative and qualitative evaluation of the drought indices. The positive aspects and limitations of these tools were also considered. The qualitative evaluation consisted of an analysis of drought indices and their theoretical and computational aspects. Meanwhile, the quantitative evaluation dealt with the indices' performance in identifying droughts in the study area.

## 3. Results

The results were divided into two parts. First, an analysis of drought events and index values was carried out, taking into consideration the Target Levels, for the period from 2002 to 2020. The second part involved assigning values for the weights of the criteria (robustness, treatability, transparency, sophistication, extensibility, and dimensionality), which were defined according to their relative importance to the drought indices. Scores were also established for the main characteristics of the indexes, which were defined based on the analysis and performance of each tool. As a result, it was possible to list the best indices to be used in the hydrological monitoring of the study area.

### 3.1. Indices Analysis in Relation to the Target Levels

From the comparison of the values of the annual volumes of the Castanhão, Orós, and Banabuiú reservoirs with the Target Levels, it was possible to quantify and categorize the drought events that occurred in the period from 2002 to 2020 in each system. Based on this identification, a detailed analysis of the index values and the drought events they are able to identify was carried out in order to represent the hydric state of the reservoirs analyzed.

#### 3.1.1. Index Analysis

The analysis of the indices, in relation to droughts, aimed to capture the sensitivity of each index in representing—or not—the drought state of the reservoirs. To this end, all index values were compared with the ratings of the Target Levels in order to quantify the events that each one was able to capture.

Tables 10–12 show the reference dates for the index calculation, as well as the volumes of the reservoirs in that period. In addition, the ratings of the Target Levels and the index values with the colors corresponding to the Target Levels are presented.

For the Banabuiú reservoir (Table 10), Target Level 1 corresponded to the years 2004 to 2006 and 2008 to 2012, having been well captured by SWSI—12 and RDI. The other indexes had difficulties in representing this event. Target Level 2 was portrayed by the years 2007 and 2013, and only SPI—12, SPI—36, SDI—12, and SDI—36 were able to capture this event. Target Levels 3 and 4 were evidenced in 2002 to 2003 and 2020, respectively, where no index was able to identify them. Target Level 5 is equivalent to the period from 2014 to 2019, being captured at times by the SWSI—12, RDI, and SPI—36.

**Table 10.** Analysis of the indexes in relation to the Target Levels for the Banabuiú reservoir.

| Date | Volume (hm$^3$) | Target Level | SWSI—12 | RDI | SDI—12 | SDI—36 | SPI—12 | SPI—36 | EDDI—12 |
|---|---|---|---|---|---|---|---|---|---|
| 31 December 2002 | 383.5 | 3 | 0.846 | −1.151 | −0.396 | −0.472 | 0.320 | −0.279 | 0.846 |
| 31 December 2003 | 374.3 | 3 | 0.795 | 0.409 | 1.177 | 1.235 | 0.014 | −0.383 | 0.795 |
| 31 December 2004 | 1213.32 | 1 | 3.560 | 2.870 | −0.309 | −0.355 | 1.314 | 0.779 | 3.560 |
| 31 December 2005 | 938.85 | 1 | 1.902 | 5.040 | 0.060 | 0.083 | −0.554 | 0.402 | 1.902 |
| 31 December 2006 | 665.23 | 1 | 0.250 | 2.473 | −0.126 | −0.241 | −0.486 | 0.182 | 0.250 |
| 31 December 2007 | 478.03 | 2 | 0.242 | 1.551 | 0.474 | 0.475 | −0.529 | −0.866 | 0.242 |
| 31 December 2008 | 935.73 | 1 | 2.525 | 2.988 | 1.104 | 1.051 | 0.126 | −0.512 | 2.525 |
| 31 December 2009 | 1216.08 | 1 | 3.345 | 0.888 | −1.011 | −0.914 | 0.999 | 0.275 | 3.345 |
| 31 December 2010 | 895.95 | 1 | 1.017 | 3.475 | 1.698 | 1.766 | −0.928 | 0.120 | 1.017 |
| 31 December 2011 | 1209.64 | 1 | 2.806 | 0.704 | −2.046 | −1.936 | 0.898 | 0.524 | 2.806 |
| 31 December 2012 | 732.93 | 1 | −0.440 | 3.104 | −1.585 | −1.606 | −2.238 | −0.881 | −0.440 |
| 31 December 2013 | 398.77 | 2 | −0.662 | 1.475 | −0.551 | −0.544 | −1.019 | −0.921 | −0.662 |
| 31 December 2014 | 55.13 | 5 | −1.961 | −0.976 | −1.647 | −1.492 | −1.085 | −2.169 | −1.961 |
| 31 December 2015 | 10.65 | 5 | −3.454 | −3.458 | −1.196 | −1.207 | −1.322 | −1.782 | −3.454 |
| 31 December 2016 | 9.01 | 5 | −3.431 | −2.640 | −0.498 | −0.489 | −1.209 | −1.878 | −3.431 |
| 31 December 2017 | 10.11 | 5 | −2.988 | −2.736 | −0.346 | −0.291 | −0.785 | −1.720 | −2.988 |
| 31 December 2018 | 63.88 | 5 | −0.761 | −2.788 | 0.501 | 0.512 | −0.356 | −1.228 | −0.761 |
| 31 December 2019 | 71.92 | 5 | −2.496 | −3.028 | 0.141 | 0.031 | −0.199 | −0.740 | −2.063 |
| 31 December 2020 | 117.00 | 4 | −0.449 | −3.317 | −0.536 | −0.529 | 0.396 | −0.148 | −0.449 |

The Castanhão reservoir (Table 11) presented Target Level 1 in the years 2004 to 2006 and 2008 to 2012. These years were identified by SWSI—12 and RDI; the other indexes presented limitations to capture these events. Target Level 2 was displayed in the years 2007 and 2013, and only SPI—12, SPI—36, SDI—12, SDI—36, and EDDI—12 were able to identify these events. Target Level 3 was observed in 2014, being captured only by EDDI—12. Target Level 4 corresponded to the years 2015 and 2020 and was identified by SWSI—12, SDI—12, SDI—36, SPI—12, and EDDI—12. Target Level 5 appeared in the period from 2002 to 2003 and 2016 to 2019 and was captured by SWSI—12, RDI, and SPI—36.

**Table 11.** Analysis of the indexes in relation to the Target Levels for the Castanhão reservoir.

| Date | Volume (hm$^3$) | Target Level | SWSI—12 | RDI | SDI—12 | SDI—36 | SPI—12 | SPI—36 | EDDI—12 |
|---|---|---|---|---|---|---|---|---|---|
| 31 December 2002 | 273.6 | 5 | −1.531 | −4.961 | −0.526 | −0.508 | 0.023 | −0.260 | 0.58 |
| 31 December 2003 | 317.56 | 5 | −2.499 | −3.807 | 0.917 | 0.916 | −0.41 | −0.786 | 0.982 |
| 31 December 2004 | 4431.67 | 1 | 3.217 | 1.9 | −1.208 | −1.181 | 1.332 | 0.496 | 0.63 |
| 31 December 2005 | 3796.44 | 1 | 0.901 | 4.53 | 0.142 | 0.152 | −0.919 | 0.055 | 0.482 |
| 31 December 2006 | 4039.16 | 1 | 2.12 | 2.915 | −0.619 | −0.600 | 0.298 | 0.413 | 0.916 |
| 31 December 2007 | 3424.3 | 2 | 1.093 | 2.471 | 0.619 | 0.622 | −0.482 | −0.628 | 0.853 |
| 31 December 2008 | 5337.28 | 1 | 3.382 | 4.042 | 1.204 | 1.199 | 1.332 | 0.602 | 0.683 |
| 31 December 2009 | 5293.13 | 1 | 3.21 | 1.359 | −0.882 | −0.859 | 0.981 | 0.970 | 0.737 |
| 31 December 2010 | 4164.12 | 1 | 0.303 | 3.46 | 1.621 | 1.611 | −0.567 | 0.933 | 1.13 |
| 31 December 2011 | 4932.84 | 1 | 2.727 | 1.03 | −1.695 | −1.661 | 1.043 | 0.764 | 0.794 |
| 31 December 2012 | 3725.88 | 1 | −0.463 | 3.204 | −0.950 | −0.926 | −2.026 | −0.664 | 1.54 |
| 31 December 2013 | 2696.94 | 2 | −0.334 | 2.143 | −0.034 | −0.022 | −0.489 | −0.624 | 1.214 |
| 31 December 2014 | 1728.09 | 3 | −0.384 | 0.865 | −1.658 | −1.625 | −0.424 | −1.550 | 1.053 |
| 31 December 2015 | 744.15 | 4 | −2.579 | −0.847 | −1.605 | −1.573 | −1.669 | −1.397 | 2.281 |
| 31 December 2016 | 341.54 | 5 | −3.069 | −1.803 | −0.923 | −0.900 | −1.161 | −1.761 | 1.909 |
| 31 December 2017 | 178.7 | 5 | −3.172 | −2.561 | −0.548 | −0.530 | −0.874 | −2.025 | 1.414 |
| 31 December 2018 | 286.62 | 5 | −1.027 | −3.456 | 0.404 | 0.410 | 0.197 | −1.001 | 1.307 |
| 31 December 2019 | 187.37 | 5 | −1.937 | −4.471 | 0.209 | 0.307 | −0.441 | −0.644 | 1.696 |
| 31 December 2020 | 750.06 | 4 | 0.197 | −3.904 | −0.505 | −0.487 | 0.782 | 0.238 | 1.696 |

The Orós reservoir (Table 12) presented the Target Level 1 in the years 2004, 2006, and 2008 to 2012, being identified by SWSI—12 and RDI; the other indexes captured few events of hydric normality. Target Level 2 was observed in the years 2005, 2007, 2013, and 2014, being captured at times by SDI—12, SDI—36, SPI—12, SPI—36, and EDDI—12. Target Level 3 appeared only in 2015 and was identified only by SPI—12. Target Level 4, on the other hand, appeared in the years 2002, 2003, 2016, and 2020, being picked up by SWSI—12,

RDI, SPI—36, and EDDI—12. The period from 2017 to 2019 was marked by Target Level 5, being identified by SWSI—12, RDI, SPI—36, and EDDI—12.

**Table 12.** Analysis of the indexes in relation to the Target Levels for the Orós reservoir.

| Date | Volume (hm³) | Target Level | SWSI—12 | RDI | SDI—12 | SDI—36 | SPI—12 | SPI—36 | EDDI—12 |
|---|---|---|---|---|---|---|---|---|---|
| 31 December 2002 | 194.97 | 4 | −2.554 | −4.423 | −0.655 | −0.631 | −0.503 | −0.685 | 0.794 |
| 31 December 2003 | 313.08 | 4 | −0.599 | −2.668 | 0.469 | 0.501 | −0.173 | −0.952 | 0.982 |
| 31 December 2004 | 1417.02 | 1 | 3.583 | 2.720 | −1.151 | −1.018 | 1.282 | 0.307 | 0.580 |
| 31 December 2005 | 995.12 | 2 | 0.396 | 4.012 | −0.177 | −0.150 | −0.763 | 0.198 | 0.530 |
| 31 December 2006 | 1060.99 | 1 | 0.876 | 2.015 | −0.812 | −0.789 | −0.087 | 0.238 | 0.916 |
| 31 December 2007 | 990.92 | 2 | 1.895 | 1.899 | 0.161 | 0.191 | −0.010 | −0.492 | 1.214 |
| 31 December 2008 | 1430.63 | 1 | 3.118 | 3.427 | 0.703 | 0.737 | 0.570 | 0.166 | 0.630 |
| 31 December 2009 | 1484.09 | 1 | 3.201 | 0.795 | −0.257 | −0.248 | 0.686 | 0.550 | 0.683 |
| 31 December 2010 | 1185.48 | 1 | 0.618 | 2.957 | 1.657 | 1.698 | −0.422 | 0.368 | 0.853 |
| 31 December 2011 | 1500.88 | 1 | 2.158 | 0.773 | −0.957 | −0.796 | 0.889 | 0.538 | 0.482 |
| 31 December 2012 | 1179.3 | 1 | 0.078 | 3.023 | −1.516 | −1.545 | −1.591 | −0.483 | 1.696 |
| 31 December 2013 | 775.8 | 2 | −1.052 | 1.416 | −0.287 | −0.260 | −1.310 | −0.883 | 1.414 |
| 31 December 2014 | 732.2 | 2 | −0.841 | 0.968 | −1.203 | −1.169 | −0.672 | −1.832 | 1.307 |
| 31 December 2015 | 468.49 | 3 | −1.974 | −0.120 | −0.947 | −0.920 | −1.364 | −1.728 | 1.909 |
| 31 December 2016 | 277.7 | 4 | −2.771 | −0.464 | −0.843 | −0.886 | −1.210 | −1.681 | 1.540 |
| 31 December 2017 | 119.71 | 5 | −3.468 | −2.671 | −0.902 | −0.879 | −1.445 | −2.079 | 1.130 |
| 31 December 2018 | 111.47 | 5 | −2.814 | −3.510 | 0.465 | 0.497 | 0.051 | −1.305 | 1.053 |
| 31 December 2019 | 101.04 | 5 | −1.693 | −4.374 | −0.263 | −0.134 | −0.571 | −0.999 | 2.281 |
| 31 December 2020 | 404.9 | 4 | 0.307 | −3.536 | −1.001 | −0.979 | 0.712 | 0.054 | 2.281 |

### 3.1.2. Quantification of Drought Events according to Their Severity

Table 13 includes the drought events quantification that each index was able to identify, according to severity. SWSI—12 and RDI were able to detect all the normality events that occurred in the three reservoirs (as specified in Table 8). Regarding Target Level 2, SPI—36 was the index that presented a greater capture of these episodes. The events occurring at Target Level 3 were identified only by SPI—12 and EDDI—12. Target Level 4 was captured, at times, by all the indices. However, SWSI—12 performed better in this distinction. Target Level 5 was identified by SWSI—12, RDI, SPI—36, and EDDI—12.

**Table 13.** Occurrence of drought events according to their severity. B, C, and O represent Banabuiú, Castanhão, and Orós, respectively.

| Target Level | SWSI—12 | | | RDI | | | SDI—12 | | | SDI—36 | | | SPI—12 | | | SPI—36 | | | EDDI—12 | | |
|---|---|---|---|---|---|---|---|---|---|---|---|---|---|---|---|---|---|---|---|---|---|
| | B | C | O | B | C | O | B | C | O | B | C | O | B | C | O | B | C | O | B | C | O |
| 1 | 8 | 8 | 7 | 8 | 8 | 7 | 3 | 3 | 2 | 3 | 3 | 2 | 3 | 4 | 4 | 2 | 5 | 2 | 2 | 1 | 1 |
| 2 | 0 | 0 | 1 | 0 | 0 | 0 | 1 | 1 | 2 | 1 | 1 | 2 | 1 | 2 | 3 | 2 | 2 | 3 | 0 | 1 | 1 |
| 3 | 0 | 0 | 0 | 0 | 0 | 0 | 0 | 0 | 0 | 0 | 0 | 0 | 0 | 0 | 1 | 0 | 0 | 0 | 0 | 1 | 0 |
| 4 | 0 | 1 | 2 | 0 | 0 | 1 | 0 | 1 | 0 | 0 | 1 | 0 | 0 | 1 | 0 | 0 | 0 | 1 | 0 | 1 | 1 |
| 5 | 2 | 2 | 1 | 2 | 4 | 2 | 0 | 0 | 0 | 0 | 0 | 0 | 0 | 0 | 0 | 1 | 1 | 1 | 0 | 0 | 1 |
| Total of Drought Events | 2 | 3 | 4 | 2 | 4 | 3 | 1 | 2 | 2 | 1 | 2 | 2 | 1 | 3 | 4 | 3 | 3 | 5 | 0 | 3 | 3 |

Consequently, out of the total 11 drought events for Bananabuiú, SPI—36 was able to record 3, followed by SWSI—12 and RDI, which were able to identify 2, and lastly, SDI—12, SDI—36, and SPI—12, which were only able to capture 1. In this same context, Castanhão also exhibited 11 years of drought, of which 4 years were identified by RDI, 3 years by SWSI—12, SPI—12, SPI—36, and EDDI—12, and 2 years by SDI—12 and SDI—36.

The Orós reservoir exhibited a 12-year drought and the index that had the greatest ability to identify these events was SPI—36, which captured five episodes. The SWSI—12 and SPI—12 were able to register four events, followed by RDI and EDDI—12 with three events, and finally, SDI—12 and SDI—36 with only two events.

### 3.1.3. Quantification of Drought Events Regardless of the Severity

A survey was carried out on the ability of each index to identify drought, regardless of severity (Table 14), i.e., at how many times did the index indicate that it was or was not in drought, even if this did not correspond to magnitude of the Target Levels.

**Table 14.** Occurrence of drought events independent of their severity. The abbreviations (B, C, and O) represent Banabuiú, Castanhão, and Orós, respectively.

| Quantity of Drought Events | | | SWSI—12 | | | RDI | | | SDI—12 | | | SDI—36 | | | SPI—12 | | | SPI—36 | | | EDDI—12 | | |
|---|---|---|---|---|---|---|---|---|---|---|---|---|---|---|---|---|---|---|---|---|---|---|---|
| B | C | O | B | C | O | B | C | O | B | C | O | B | C | O | B | C | O | B | C | O | B | C | O |
| 11 | 11 | 12 | 5 | 7 | 7 | 7 | 7 | 6 | 7 | 7 | 9 | 7 | 7 | 9 | 11 | 10 | 11 | 11 | 11 | 12 | 2 | 11 | 12 |

Of the 11 drought events registered for Banabuiú reservoir, the indexes that managed to identify the most episodes was the SPI—12 and the SPI—36 (11 events), followed by the SDI—12 and SDI—36 (7 events), RDI (7 events), SWSI—12 (5 events), and EDDI—12 (2 events).

As for the Castanhão reservoir, there is the SPI—36 and EDDI—12 with the identification of 11 events, followed by SPI—12 with 10 events and SWSI—12, RDI, SDI—12, and SDI—36 with 7 episodes each.

For the Orós reservoir, the SPI—36 and EDDI—12 indices were able to identify the 12 drought episodes. While the SPI—12 captured 11 events, the SDI—12 and SDI—36 identified 9 events each, and the SWSI—12 7 and the RDI identified 6 events each.

### 3.2. Comparative Evaluation of Drought Indices

#### 3.2.1. Robustness

Robustness was chosen as the most important criterion and received the maximum score regarding the weights, i.e., value of 8 and relative importance of 27%. SPI and EDDI were quite responsive in detecting drought conditions in the study region, where the former was very sensitive to precipitation variations and the latter to potential evapotranspiration variations. However, these indices do not take into account the variability of water resources within the basin. Therefore, for the robustness criterion, SPI and EDDI received a value of 4.

The SWSI and RDI are hydro-meteorological indexes, in which the former uses three input variables (precipitation, affluent streamflow, and reservoir volume) in its composition and the latter uses four (precipitation, affluent streamflow, volume, and temperature). This range of variables allows the indexes to analyze not only factors related to precipitation but also to water availability in the reservoirs. Thus, the SWSI and the RDI showed good capture of drought according to its severity, managing to identify, at some times, more drought episodes than the SPI. However, regardless of severity, these indices tend to capture less than 65% of the occurred episodes when analyzing the occurrence of drought events. For this reason, the score assigned to the two indices for the robustness criterion was 3. The SDI, on the other hand, presented a lower identification of drought in relation to its magnitude but exhibited a good capture of the events regardless of severity, so the robustness score for it was 3.

#### 3.2.2. Treatability

The treatability received the weight of 6 (20% of relative importance), because the institutions responsible for monitoring droughts in Brazil, such as FUNCEME, tend to opt for more treatable indices, as these are easier to be implemented and generated. Regarding the treatability criterion, the indices were evaluated in relation to the ease of calculation (number of steps) and the required input variables.

Thus, SPI obtained the highest score (4) in relation to the other indexes since it uses precipitation data only in its formulation (easily accessible data) and it presents three calculation steps.

Although the EDDI requires only PET data and displays, like the SPI, three calculation steps, the data available for calculating the index have a more restricted access and often flaws in its construction. For these reasons, a rating of 3 has been assigned to the EDDI.

SDI, like SPI and EDDI, requires only one input variable, which in this case is the streamflow. SDI calculation is more complex and covers five steps, so SDI was given a score of 3.

SWSI and RDI are more complex to calculate and have more input variables involved. However, RDI needs one more variable (air temperature near to surface) and its calculation has more steps when compared to SWSI. Thus, SWSI received a score of 3 and RDI a score of 2 for the treatability criterion.

### 3.2.3. Transparency

Transparency was given a weight of 5 (17% of relative importance), as the indices used in drought monitoring are expected to be easily understood by the general public. In this way, the indexes can help, for example, farmers to define the best time to plant, or the managers of water resources to determine the moment of release or storage of water in reservoirs.

Essentially, the indexes presented in this paper are easy to understand by researchers and professionals in the area but are not well understood by the general public. Therefore, SPI and SDI received a score of 3, and the other indexes (SWSI, RDI, and EDDI), for being more complex, received a score of 2.

### 3.2.4. Sophistication

Since the goal of this work is to identify the indices that can be used for hydrological monitoring of the Castanhão, Banabuiú, and Orós reservoirs, and this identification is related to a series of hydro-meteorological factors, the use of more sophisticated tools is necessary. However, one of the disadvantages of more complex approaches to identify droughts is that they usually require greater availability and quality of data, which makes them less transparent and less tractable. In addition, indices that have a greater number of input parameters tend to be more sophisticated, so this variety of parameters allows the index to better assess the conditions that influence drought events.

Thus, the weight of 6 was assigned to the sophistication criterion, which has a relative importance of 20%. The SWSI and RDI indices received the highest scores (5) because they require a greater number of input data. Their calculations are based on hydrometeorological variables, and both exhibit the ability to identify drought events according to their magnitude. As for the other indices, SPI, SDI, and EDDI, a score of 3 was given, as they are less sophisticated indices when compared to SWSI and RDI.

### 3.2.5. Extensibility

Extensible indices present greater importance for decision makers, as they devise action plans based on previous droughts. In this case, extensibility received a weight of 3 (10% relative importance), because its relevance is lower when compared to the previously mentioned criteria. So, it was considered more important the index's ability to identify droughts, its degree of sophistication, than if it were easy to understand and with a simpler calculation.

Precipitation data series is long (more than 40 years), which allows the indexes that depend on this variable to analyze the behavior of droughts in the past and identify behavior trends for these events. On the other hand, the affluent streamflow can be estimated by rainfall/flow hydrological models [56], which are calibrated by means of variables such as precipitation, so that the series obtained are simulated and not observed. In this case, one can have a generation of very extensive hydrological series, thus allowing the index expansion.

The reservoir volume (water level) data are limited to the beginning of reservoir operation, which hinders the extensibility of the index, since many reservoirs were built

recently, as is the case of the Castanhão (2002). As an alternative for the extension of these data, there is an equivalent reservoir approach, which aims to reproduce the characteristics of that body of water in order to simulate volume and level data. However, this is not a widespread technique among data provider institutions, so they end up providing only the series of volumes collected after the beginning of reservoir activities. Thus, even if it is possible to obtain extensive precipitation and streamflow series, the volume data would end up limiting the period for calculating the indices.

The same analysis can be performed for potential evapotranspiration data, since they need the temperature to be calculated, as in the method from [57], or other components (wind speed, insolation, and relative humidity) to use the Penman–Monteith estimation. This need for other variables ends up limiting the size of the potential evapotranspiration series because not all rainfall stations have sensors to detect local temperature or collect the other necessary data.

Based on the aforesaid, the SPI received the highest extensibility score (5), for being an index that uses only precipitation as an input variable, which enables expansion and, consequently, a better analysis of past droughts. The SDI was assigned a score of 3 because it uses streamflow in its equation and presents limitations regarding these data since they come from hydrological models.

SWSI and EDDI received the same scores (3), which are justified by the fact that SWSI presents more input variables, which can directly impact the expansion of its values since the reservoir volume is a limiting data. Concerning EDDI, the evapotranspiration data may present flaws or even be absent, thus making it impossible to expand the index.

RDI was the index that received the lowest score (2) because besides using the same data as the SWSI, it also incorporates temperature, which can be limiting data for the expansion of this index to the regions.

### 3.2.6. Dimensionality

For the dimensionality criterion, it is desirable that the index has a simple unit with physical meaning, such as $m^3$ of water volume and percentage of rainfall, rather than dimensionless or complex units [8], to allow the index to connect clearly with the physical conditions of the environment. Thus, simpler indices such as standardized anomalies and percentiles are advantageous for comparing resources across locations and time periods.

In this case, weight of 2 (6% relative importance) was assigned to this criterion, since part of the drought indices discussed in this paper exhibit more complex or dimensionless units. Regarding the scores, the indices SPI, SDI, and EDDI received the highest values (4), which are justified by their simplicity and the fundamental units they represent. Meanwhile, the SWSI and RDI received a score of 3, as they are more complex indices that display dimensionless information.

### 3.2.7. Analysis of the Results between the Indexes and the Decision Criteria

Based on the qualitative and quantitative evaluations, this study points out that SPI is better than SWSI, EDDI, SDI, and RDI for quantifying drought conditions in the Banabuiú, Castanhão, and Orós reservoirs. Thus, the total scores assigned to the indices were 118 for SPI, 97 for SWSI, 95 for EDDI and SDI, and 88 for RDI (Table 15).

**Table 15.** Qualitative evaluation of drought indexes.

| Characteristics | Weights | Relative Importance (%) | SPI | SWSI | EDDI | SDI | RDI |
|---|---|---|---|---|---|---|---|
| Robustness | 8 | 27 | 4 | 3 | 4 | 3 | 3 |
| Sophistication | 6 | 20 | 4 | 5 | 3 | 3 | 5 |
| Treatability | 6 | 20 | 4 | 3 | 3 | 3 | 2 |
| Transparency | 5 | 17 | 3 | 2 | 2 | 3 | 2 |
| Extensibility | 3 | 10 | 5 | 3 | 3 | 4 | 2 |
| Dimensionality | 2 | 6 | 4 | 3 | 4 | 4 | 3 |
| Total Points | 30 | 100 | 118 | 97 | 95 | 95 | 88 |

The result regarding SPI has already been pointed out in previous works, such as those from [8,55,58], which determined that this index was one of the most suitable to be used in the monitoring of meteorological drought.

Within this context, the SPI ranked well on all six decision criteria because it has a good ability to measure drought over a wide range of conditions and can be calculated for various scales of interest (monthly, quarterly, semiannual, annual, biennial, and triennial), is spatially and temporally comparable, and has a simple calculation. Moreover, it uses only one input variable (precipitation), its values are easily understood by the scientific community (positive values indicate wetter than normal conditions and negative values indicate drier than normal conditions), its time series can be extended, and the index values can be compared to fundamental units. However, Quirind (2009) points out in his paper that SPI has limitations in arid locations, which exhibit seasons with no precipitation.

The SWSI has also been evaluated in relation to the criteria by the researchers [17,55], where the former attributed a higher score to the SWSI in relation to the SPI because it was considered more robust and sophisticated. On the other hand, the second study pointed out that the SPI is superior to the SWSI in the six criteria used, and therefore attributed a lower score to the SWSI.

Thus, when compared to the other indexes, the SWSI was identified as the second best to identify droughts in the reservoirs under study. It showed a good capacity to identify drought episodes according to magnitude, however, it showed limitations to capture these events despite the severity. The SWSI can also be calculated for various time spaces, being spatially and temporally comparable, but it presents a greater complexity of calculation, with more variables involved, which makes it difficult for users to understand. Its series can be extended; however, the volume data are a limiting factor, and its values are dimensionless.

EDDI and SDI ranked third, and both received the same score, differing in the criteria of robustness, transparency, and extensibility. First of all, EDDI proved to be more robust than SDI, as it was able to better capture the drought conditions of the reservoirs. Regarding transparency, EDDI obtained a lower evaluation than SDI, because it presents a more complex calculation methodology, and more difficult to understand by final users. In the extensibility criterion, EDDI also obtained a lower score, because the calculation of potential evapotranspiration needs other variables, which may limit the index expansion.

RDI obtained the same scores for robustness, transparency, sophistication, and dimensionality as SWSI, because they are similar indexes in construction and input variables. The criteria for differentiation between them were treatability and extensibility, as the RDI presents one more variable than the SWSI and a more complex calculation methodology. In addition, the RDI presents more restrictions to be extended, considering that it uses two variables (volume and temperature), which present limitations in obtaining and extending.

## 4. Conclusions

The present article analyzed the performance of five drought indices (SPI, SWSI, EDDI, SDI, and RDI), with the objective of identifying the most suitable index for the hydrological monitoring of the Castanhão, Orós, and Banabuiú reservoirs, located in the state of Ceará, Brazil.

Thus, it was found that the 36-month SPI was able to better represent drought episodes, identifying a greater number of events, when compared to the 12-month SPI and the other analyzed indices. Moreover, the SPI received the highest score related to the evaluation criteria, as it showed a good ability to identify drought, being a tool of easy application and understanding, with a relatively simple calculation, and can also be compared to fundamental units. Therefore, the SPI is able to be easily employed and operationalized in systems of monitoring and prevention against extreme drought events.

The SWSI and the RDI showed good identification of drought events according to severity; however, when they were analyzed independently of the severity, both showed limitations. Regarding the decision criteria analysis, the SWSI received a higher score

than the RDI, because its calculation involves fewer variables, and its series can be more easily expanded in time. In this case, the SWSI would be the most suitable for monitoring hydrological drought in the study region.

The EDDI was able to identify most drought events both according to the magnitude and independent of it, showing sensitivity to the variation of potential evapotranspiration. Moreover, it is an index that can be used to represent fundamental units. However, its calculation complexity and the difficulty of obtaining data end up limiting the use and implementation of this index in monitoring systems.

On the other hand, the SDI was the one that could identify fewer drought events according to their severity. However, it is a relatively simple index to calculate and easily understood by users. Moreover, its values can be temporally expanded, and its results can be compared with fundamental units. However, its limitations regarding robustness and sophistication have made it less suitable for hydrological drought monitoring when compared to the SWSI.

The indices that best captured drought events based on hydro-meteorological monitoring in the studied reservoirs and based on the decision criteria scores are, in descending order of score, SPI (118), SWSI (97), EDDI (95), SDI (95), and the RDI (88).

The SPI is already a consolidated index for at the drought monitoring, composing several systems, e.g., the Drought Monitor of the United States, Germany, and Brazil. However, the SWSI is not as widespread as the SPI. Thus, further studies applying the SWSI are suggested, in order to adapt it to the different regions and use it. This index considers more wide drought conditions, since it considers hydrological and meteorological factors in its composition, as well as water level variability of the reservoirs.

**Author Contributions:** Conceptualization, methodology and validation, S.T.N.G. and F.d.C.V.J.; software, S.T.N.G. and F.d.C.V.J.; formal analysis, investigation, resources, and data curation, S.T.N.G., F.d.C.V.J., and C.d.S.S.; writing—original draft preparation, S.T.N.G., F.d.C.V.J., and C.d.S.S.; writing— review and editing, visualization, and supervision, S.T.N.G., F.d.C.V.J., C.d.S.S., D.A.C.C., E.S.P.R.M., and J.M.F.d.C. All authors have read and agreed to the published version of the manuscript.

**Funding:** This research was funded by the National Council for Scientific and Technological Development (CNPQ), process number 409666/2021-1 and the APC was funded by the Cearense Foundation for Scientific and Technological Development (FUNCAP), process number PS 1-0186—00326.01.00/21.

**Data Availability Statement:** The data used in this research came from FUNCEME (precipitation, streamflow and potential evapotranspiration), COGERH (level and volume of reservoirs) and INMET (temperature), which are listed below (all accessed on 12 January 2023):

Precipitation data. Available at: http://www3.funceme.br/web/storage/obs/modhydro/inmet-ana-sinda-estados/eixo_norte/incrementais/asc//.
Streamflow data. Available at: http://www3.funceme.br/web/storage/obs/modhac/qca/reservatorios_semiarido/.
Potential evapotranspiration data. Available at: http://www3.funceme.br/web/storage/obs/interpolation_kriging_evapotranspiration/monthly_files/.
Reservoir level and volume data. Available at: http://www.hidro.ce.gov.br.
Temperature data. Available in: https://bdmep.inmet.gov.br.

**Acknowledgments:** The authors thank the National Council for Scientific and Technological Development (CNPQ) and Cearense Foundation for Scientific and Technological Development (FUNCAP) for all the support for this research.

**Conflicts of Interest:** The authors declare no conflict of interest.

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
