# Peer review of "Comparative Analysis of Drought Indices in Hydrological Monitoring in Ceará’s Semi-Arid Basins, Brazil"

_water, doi:10.3390/w15071259_

Round 1

Reviewer 1 Report

Figure 1 was not called out in the text

The authors need to make a contingency table or something of the sort, explaining what the percentage of data failure is, method used for filling in, and show the posts used on a map, so you can get an idea of their spatial distribution. 

They need to explain and identify all the terms that appear in the equations of the indexes. 

It was not very clear why the target reservoir methodology was used to identify which would be the best method for calculating the drought index. If it is the reference why calculate indexes.  This needs to be well explained in the text.

It is not clear how the authors arrived at Table 8. What statistical criteria were used? 

The article needs to make clearer what is the great gain of using an index for drought monitoring, in detriment to the direct analysis of the time series of hydrological and meteorological variables. 

Finally, the SPI index is already a well studied and applied index. What would then be the great scientific contribution of the work, besides reaffirming the use of an already established index?

Author Response

REVISOR 1

Os autores implementaram alterações no artigo para atender às sugestões e recomendações do revisor. O texto do manuscrito foi revisado novamente para a língua inglesa. Este documento foi alterado, as alterações são indicadas por página, número de linha e destaques amarelos. Os roteiros matemáticos do artigo foram utilizados de acordo com as recomendações da revista.

1. A Figura 1 não foi destacada no texto .

A frase foi corrigida com a citação da Figura 1.

2. Os autores precisam fazer uma tabela de contingência ou algo do tipo, explicando qual é o percentual de falha dos dados, forma de preenchimento, e mostrar as postagens utilizadas em um mapa, para você ter uma ideia de sua distribuição espacial . 

The data used did not present any gaps in its time series. A map containing the study area and the spatial distribution of the rainfall stations used in the work was prepared and inserted as Figure 2.

 3. They need to explain and identify all the terms that appear in the equations of the indexes. 

The equations have been reviewed and corrected.

4. It was not very clear why the target reservoir methodology was used to identify which would be the best method for calculating the drought index. If it is the reference why calculate indexes. This needs to be well explained in the text.

The target levels were used in order to identify the years with events with years of scarcity or water abundance in the reservoirs, as they are directly related to the availability of water in each system and also in function of the demand and even with the indication of the levels for the operation of these reservoirs.

Drought indices are tools that use, individually or combined, climatological and/or hydrological variables in their composition, thus generating indicators, which will be used for monitoring and characterizing drought events both in terms of their severity, duration, and identification. of the impacts. Different drought indices have the potential to be used for “preparedness for action” and also associated with triggers, this approach is associated with one of the pillars of Proactive Drought Management: Monitoring and early warning.

In this study, we evaluated some indices with a view to investigating their performance in terms of identifying impacts of water scarcity in reservoirs in the semi-arid region of Brazil. In fact, measurements of the level of reservoirs could be used as a drought index or as an indicator, however many reservoirs have monitoring gaps, or sparse measurements, in addition, others do not even have monitoring, mainly those that supply smaller cities and has broader social impact for agriculture and animal watering.

5. It is not clear how the authors arrived at Table 8. What statistical criteria were used? 

Table 8 was made from the comparison between the target levels and the actual levels of the reservoirs studied during the period analyzed (2002 to 2020). In this analysis, it was admitted that each year would correspond to an event and the occurrence of the target levels was estimated in a percentage form. For instance, for the year 2002, Banabuiú and Castanhão reservoirs presented 8 years of normality (Target Level 1), corresponding to 42.11% of the events. The Orós reservoir, on the other hand, had 7 years of normality, with a percentage of 36.84%. For the Target Level 2, the Banabuiú and Castanhão reservoirs accounted for 2 events, representing 10.53%, while Orós showed 4, equivalent to 21.05%. For the Target Level 3, Banabuiú showed 2 events (10.53%), and Castanhão and Orós showed 1 (5.26%). Goal Level 4 had 1 event (5.26%) in Banabuiú, 2 events (10.53%) in Castanhão, and 4 events (21.05%) in Orós. The Target Level 5, presented 6 events (31.58%) in Banabuiú and Castanhão, and only 3 (15.79%) in the Orós reservoir.

6. The article needs to make clear what the big gain is in using an index for drought monitoring, as opposed to directly analyzing the time series of hydrological and meteorological variables.

TThe great gain in using drought indices is that they reflect the effects associated with water scarcity, whether meteorological drought, water drought or agricultural drought.

Above all, this manuscript also evaluates the adaptation applied to drought indices that were integrated for other regions and contexts around the world, such as the SWSI and RDI. In general, these indices are composed of several variables, so that their information becomes much more comprehensive when detected only using the target levels. Still in this context, some of the indices used, as in the case of the SWSI and RDI, have predictive potential (Shafer; Dezman, 1982; Garen, 1993; Weghorst, 1996), which allows identifying trends in the behavior of droughts, helping to take decision by managers.

The analysis of the comparison of the values of the indices with the target levels of the reservoirs enables us to identify whether or not the indices represent the water reality of that hydro-system, thus validating the adaptation carried out for the study area. The evaluation is broader than just comparison based on statistical statistics, but also reflects ease of use, applicability and viability, using as a basis the methodology suggested by (Keyantash; Dracup, 2002).

7. Finally, the SPI index is already a well studied and applied index. What would then be the great scientific contribution of the work, besides reaffirming the use of an already established index?

The identification of the best drought indices for a region is a very important part of the monitoring process, since some indices may not represent the reality of the system analyzed, thus providing false information, which can hinder the development of contingency plans in the short term, or in a more strategic approach in the construction of Drought Plans, for example.

This manuscript evaluated 5 drought indices and after these analyses, it was found that the two best for employment in the region are the SPI and the SWSI. However, the SPI with a scale of 36 months was the one that presented the best results, that is, the use on longer scales is recommended to monitor hydrological droughts.

In addition to the contributions already commented in the other questions above, the discussion made here to the use of SPI on longer time scales such as 36 months for monitoring hydrological droughts is important, because in Brazil the scales of 12 to 24 months are commonly considered in operational drought monitoring (Martins et al. 2015).

Além disso, o SWSI é pouco utilizado em estudos e monitoramento operacional no Brasil, tanto pela falta de estudos para adequar os parâmetros às características climáticas da região quanto pela dificuldade de obtenção das variáveis ​​em tempo operacional para uso, como o Brasil Monitor de Secas (www.monitordesecas.ana.gov.br). Nesse sentido, este estudo amplia a discussão sobre a utilização do SWSI como ferramenta de monitoramento operacional e tem potencial preditivo das condições de seca no Brasil, uma vez que abrange fatores hidrológicos e meteorológicos em sua composição, considerando também a variabilidade hídrica dos reservatórios .

Reviewer 2 Report

Please find attached document with comments for your kind consideration. I strongly recommend that authors check the language used as several portions of the write-up lacks the proper scientific writing style. 

Author Response

REVIEWER 2

The authors implemented changes to the article to meet the suggestions and recommendations the reviewer. The manuscript text was revised again for the English language. This document has changed, changes are indicated by page, line number and yellow highlights. The mathematical scripts in the article were used in accordance with the journal's recommendations.

1. Please find attached document with comments for your kind consideration. I strongly recommend that authors check the language used as several portions of the write-up lacks the proper scientific writing style. 

The following modifications were made:

- The text of the abstract was rewritten, in order to clarify the results and conclusions of the work.

- The sentence of line 27 and 28 has been rewritten.

- There were changes in the following lines:

37 (changes for change)

61 (Deleted ‘the’)

68 (Deleted ‘you’)

74, 85, 91, 97, 149 (reference)

235 (the word has been modified)

- All references have been corrected and adapted to the journal's standard.

Reviewer 3 Report

            The manuscript entitled “Comparative Analysis of Drought Indices in Hydrological Monitoring in Ceará's Semi-arid Basins” (reference water--2190421) evaluated the performances of five drought indices used in Northeast Brazil. The paper is well organized and the topic is interesting. However, the English writing is poor, and many mistakes exist in the text, which make the manuscript not easy to be understood. Some other comments are shown as below:

Line 75 and many others: The reference (e.g., 16 and 17) format in the text is not correct.

Line 90: What is IDA? You mean ADI?

Fig.2: Words in the figure are too small to recognize.

Line158: MRF or RMF?

Line 298: Where is Vk in the equation?

Line 323: How H(x) both larger than 0 and 0.5?

Author Response

REVIEWER 3

The authors implemented changes to the article to meet the suggestions and recommendations the reviewer. The manuscript text was revised again for the English language. This document has changed, changes are indicated by page, line number and yellow highlights. The mathematical scripts in the article were used in accordance with the journal's recommendations.

1. The manuscript entitled “Comparative Analysis of Drought Indices in Hydrological Monitoring in Ceará's Semi-arid Basins” (reference water--2190421) evaluated the performances of five drought indices used in Northeast Brazil. The paper is well organized and the topic is interesting. However, the English writing is poor, and many mistakes exist in the text, which make the manuscript not easy to be understood. Some other comments are shown as below:

The manuscript was revised for English improvement.

2. Line 75 and many others: The reference (e.g., 16 and 17) format in the text is not correct.

The reference formats cited in the text, lines 74, 85 and 97, as well as the others, were corrected.

3. Line 90: What is IDA? Do you mean ADI?

The abbreviation IDA has been replaced by ADI, as it refers to the Aggregate Drought Index, on line 86.

4. Fig.2:  Words in the figure are too small to recognize.

The figure 2 was updated.

5. Line 158: MRF or RMF?

The acronym was corrected to MRF, as it refers to Metropolitan Region of Fortaleza, line 148.

6. Line 298: Where is Vk in the equation?

The equation was corrected.

7. Line 323: How H(x) both larger than 0 and 0.5?

The value of H(x) in Equation 5 is greater than 0 and less than or equal to 0.5. For Equation 6, the value of H(x) is greater than 0.5 and less than or equal to 1. The Equations have been corrected.

Round 2

Reviewer 1 Report

I understand that all questions have been answered to my satisfaction. But I believe that some answers could also have gone into the text. But that is up to the authors. 

Reviewer 3 Report

No